

# Tropospheric delay parameters from numerical weather models for multi-GNSS precise positioning

Cuixian Lu[1], Florian Zus[1], Maorong Ge[1], Robert Heinkelmann[1], Galina Dick[1], Jens Wickert[1,2], and Harald Schuh[1,2]

[1]German Research Centre for Geosciences GFZ, Telegrafenberg, 14473 Potsdam, Germany
[2]Technische Universität Berlin, Institute of Geodesy and Geoinformation Science, 10623 Berlin, Germany

*Correspondence to*: Cuixian Lu (cuixian@gfz-potsdam.de)

**Abstract.** The recent dramatic development of multi-GNSS (Global Navigation Satellite Systems) constellations brings great opportunities and potential for more enhanced precise positioning, navigation, timing, and other applications. Significant improvement on positioning accuracy, reliability, as well as convergence time with the multi-GNSS fusion can be observed in comparison with the single-system processing like GPS (Global Positioning System). In this study, we develop a numerical weather model (NWM)-constrained PPP processing system to improve the multi-GNSS precise positioning. Tropospheric delay parameters which are derived from the European Centre for Medium-Range Weather Forecasts (ECMWF) analysis are applied to the multi-GNSS PPP, a combination of four systems: GPS, GLONASS, Galileo, and BeiDou. Observations from stations of the IGS (International GNSS Service) Multi-GNSS Experiments (MGEX) network are processed, with both the standard multi-GNSS PPP and the developed NWM-constrained multi-GNSS PPP processing. The high quality and accuracy of the tropospheric delay parameters derived from ECMWF are demonstrated through





comparison and validation with the IGS final tropospheric delay products. Compared to the standard
PPP solution, the convergence time is shortened by 32.0 %, 37.5 %, and 25.0 % for the north, east, and
vertical components, respectively, with the NWM-constrained PPP solution. The positioning accuracy
also benefits from the NWM-constrained PPP solution, which gets improved by 2.5 %, 12.1 %, and
18.7 % for the north, east, and vertical components, respectively.
**Keywords.** Multi-GNSS; GPS; Tropospheric delay parameters; Numerical weather models; Precise
point positioning (PPP); Convergence time; Positioning accuracy.
**1 Introduction**
As the first space-based satellite navigation system, Global Positioning System (GPS) consisting of a
dedicated satellite constellation has been extensively applied for many geodetic applications in the last
decades (Ge et al., 2008; Li et al., 2013). In particular, the GPS Precise Point Positioning (PPP,
Zumberge et al., 1997) technique draws special interests as it enables accurate positioning of mm to cm
accuracy with a single receiver (Blewitt et al., 2006). Due to its significant advantages in terms of
operational flexibility, global coverage, cost-efficiency, and high accuracy, the PPP approach has been
demonstrated to be a powerful tool and it is widely used in various fields such as Precise Orbit
Determination (POD) of Low Earth Orbiters (LEO), crustal deformation monitoring, precise timing,
GPS meteorology, and kinematic positioning of mobile platforms (Zumberge et al., 1997; Kouba and H





éroux, 2001; Gao and Shen, 2001; Zhang and Andersen, 2006; Ge et al., 2008). With the continuously
improved density of the tracking network infrastructure as well as the enhanced precise satellite orbit
and clock correction products with short-latency (e.g., real-time) availability, many innovative
applications like geo-hazard monitoring, seismology, nowcasting of severe weather events or regional
short-term forecasting based on the PPP technique have also been emerging and undergoing great
developments (Larson et al., 2003; Li et al., 2013; Lu et al., 2015). However, the GPS-only PPP shows
limitations concerning the convergence time, positioning accuracy, and long re-initialization period due
to insufficient satellite visibility and limited spatial geometry, especially under constrained
environmental conditions where the signals are blocked or interrupted.
The world of satellite navigation is going through dramatic changes and is stepping into a stage of
multi-constellation GNSS (Global Navigation Satellite Systems) (Montenbruck et al., 2014). Not only is
GPS of full capability and under continuous modernization, but also GLONASS has finished the
revitalization and is now fully operational. Besides, two new constellations, Galileo and BeiDou, have
recently emerged. The European Galileo currently comprises of 12 satellites deployed in orbit and it is
working towards a fully operational stage. The Chinese BeiDou officially launched a continuous
positioning, navigation, and timing (PNT) service covering the whole Asia-Pacific region at the end of
2012. It is continuously developing to a global system in the near future. In addition, the Japanese
Quasi-Zenith Satellite System (QZSS) and the Indian Regional Navigation Satellite System (IRNSS)
are also growing, with one and five satellites currently (as of 2016) operating in orbit, respectively. So



far, more than 80 navigation satellites can be in view and transmit data benefitting from the
multi-constellation GNSS, which brings great opportunities for more precise positioning, navigation,
timing, remote sensing, and other applications (Ge et al., 2012).

Undoubtedly, the integration of all existing navigation satellite systems could provide more

observations and could thus enable definite improvements on reliability, positioning accuracy and
convergence time of PPP in comparison with the stand-alone GPS PPP. Li et al. (2015a) developed a
four-system (GPS+ GLONASS + Galileo + BeiDou) positioning model to fully exploit all available
observables from different GNSS. They demonstrated that the fusion of multiple GNSS showed a
significant effect on shortening the convergence time and improving the positioning accuracy when
compared to single-system PPP solutions. The benefits of the four-system model were also found when
applied for real-time precise positioning (Li et al., 2015b), where a reduction of the convergence time
by about 70 % and an improvement of the positioning accuracy by about 25 % with respect to the
GPS-only processing were illustrated. The fusion of multi-GNSS constellations has developed to be one
of the hot topics within the GNSS community, not only limited to precise positioning but also for
related applications. For example, the multi-GNSS PPP exhibits significant advantages for GNSS
meteorology applications, such as the real-time retrieval of atmospheric parameters including integrated
water vapor, tropospheric delays, and horizontal gradients, in particular for the high-temporal resolution
tropospheric gradients (Li et al., 2015c; Lu et al., 2016). Therefore, to improve the performance of
multi-GNSS precise positioning concerning both positioning accuracy and solution convergence, is the





main focus of our study.
Numerical weather models (NWM) are able to provide the required information for describing the
neutral atmosphere, from which the meteorological parameters can be derived at any location and at any
time by applying interpolation, within the area and time window considered by the model (Pany et al.,
2001). In the past, the application of NWM in space geodetic analysis mainly focused on the
determination of mapping functions (Niell, 1996; Boehm et al., 2006). With respect to the
improvements in spatiotemporal resolutions as well as in precision and accuracy of the NWM during
recent years, tropospheric delay parameters, such as zenith total delay (ZTD), slant total delay, and
tropospheric gradients, derived from the NWM could satisfy the accuracy requirements for most GNSS
applications (Andrei and Chen, 2008). Data from the NWM have been used to perform tropospheric
delay modeling or correct for the neutral atmospheric effects in GNSS data processing. Hobiger et al.
(2008a) made use of ray-traced slant total delays derived from a regional NWM for GPS PPP within the
area of Eastern Asia. They demonstrated an improvement of station coordinate repeatability by using
this strategy in comparison to the standard PPP approach where the tropospheric delays were estimated
as unknown parameters. Furthermore, an enhanced algorithm for extracting the ray-traced tropospheric
delays of higher accuracy from the NWM in real-time mode was proposed by Hobiger et al. (2008b).
The authors presented the potential and the feasibility of applying the NWM-derived tropospheric delay
corrections into real-time PPP processing. Besides, Ibrahim and El-Rabbany (2011) evaluated the
performance of implementing tropospheric corrections from the NOAA (National Oceanic and



Atmospheric Administration) Tropospheric Signal Delay Model (NOAATrop) into GPS PPP. They
pointed out an improvement of convergence time by about 1 %, 10 %, and 15 % for the latitude,
longitude, and height components, respectively, by using the NOAA troposphere model when compared
to the results achieved with the previously used Hopfield model.
In this study, we develop a NWM-constrained PPP processing method to improve the multi-GNSS
(a combination of four systems: GPS, GLONASS, Galileo, and BeiDou) precise positioning.
Tropospheric delay parameters, which are derived from the European Centre for Medium-Range
Weather Forecasts (ECMWF, http://www.ecmwf.int/) analysis are applied to multi-GNSS PPP.
Observations from the IGS (International GNSS Service) Multi-GNSS Experiments (MGEX) network
are processed. The quality of tropospheric delay parameters retrieved from the ECMWF analysis is
assessed    by    comparison    with    the    IGS    final    tropospheric    delay    products
(ftp://cddis.gsfc.nasa.gov/gnss/products/troposphere/zpd/). The performance of multi-GNSS PPP
making use of the NWM-derived tropospheric delay parameters is evaluated in terms of both
convergence time and positioning accuracy.
This article is organized as follows: Section 2 illustrates the IGS tracking network for MGEX, the
multi-GNSS data collection, and the tropospheric delay parameters retrieved from ECMWF. Two
multi-GNSS PPP processing scenarios, the standard and the NWM-constrained PPP, are presented in
detail focusing on the modeling of the tropospheric delays. Thereafter, Section 3 describes the
comparison of tropospheric delay parameters from ECMWF with respect to the IGS final tropospheric



delay products. In Sect. 3, the positioning results, in terms of the convergence time and the positioning
accuracy, achieved with the NWM-constrained multi-GNSS PPP solution are illustrated in comparison
to the ones with the standard PPP solution. The conclusions and discussions are presented in Sect. 4.

**2 Data collection and processing**
**2.1 Multi-GNSS data collection**
In response to the dramatic development of the global satellite navigation world along with the
upcoming systems and signals, the IGS initialized the MGEX campaign to enable a multi-GNSS service
of tracking, collecting, and analyzing data of all available signals from GPS, GLONASS, BeiDou,
Galileo, QZSS, and any other space-based augmentation system (SBAS) of interest (Montenbruck et al.,
2014). Accordingly, a new worldwide network of multi-GNSS monitoring stations under the framework
of the MGEX project has been deployed in the past two years in parallel with the IGS network, which
only serves for GPS and GLONASS. Currently, the MGEX network consists of more than 120 stations,
which are globally distributed and provide excellent capability of multi-GNSS constellation tracking
and data delivering owing to the contributions from about 27 agencies, universities, and other
institutions of 16 countries (http://igs.org/mgex). Besides the tracking of the GPS constellation, the
majority of the MGEX stations enable offering the GLONASS data. At least one of the new BeiDou,
Galileo, or QZSS constellations can be tracked for each MGEX station. Today, about 75 stations are
capable of tracking the Galileo satellites, 80 stations are tracking the GLONASS satellites, and the





BeiDou constellation is supported by more than 30 receivers. Figure 1 shows the geographical
distribution of the MGEX stations and their supported constellations, except GPS, which can be tracked
by each station.

**2.2 NWM data collection**
The pressure, temperature, and specific humidity fields of the ECMWF operational analysis are utilized
to retrieve the tropospheric delay parameters. The ECMWF data are available at the German Research
Centre for Geosciences (GFZ) with a horizontal resolution of $1\,°\times1\,°$ on 137 vertical model levels
extending from the Earth's surface to about 80 km. We use the ray-trace algorithm proposed by Zus et al.
(2014) and compute station specific zenith hydrostatic (non-hydrostatic) delays, derive all three
hydrostatic (non-hydrostatic) mapping function coefficients (Zus et al., 2015a) and the horizontal delay
gradient components (Zus et al., 2015b). The calculated station-specific tropospheric delay parameters
are available every six hours per day and are valid at 0, 6, 12, and 18 UTC.

**2.3 Multi-GNSS PPP processing**
In the PPP processing, precise satellite orbits and clocks are fixed to previously determined values. The
multi-GNSS (here GPS, GLONASS, Galileo, and BeiDou) PPP processing model can be expressed as
follows,






$$
\begin{cases}
l_{r,j}^{G} = -\mathbf{u}_{r}^{G}\cdot\mathbf{r} + t_r + \lambda_{jG}(b_{rG,j}-b_{j}^{G}) + \lambda_{jG}N_{r,j}^{G} - \kappa_{jG}\cdot I_{r,1}^{G} + T + \varepsilon_{r,j}^{G} \\
l_{r,j}^{R_k} = -\mathbf{u}_{r}^{R}\cdot\mathbf{r} + t_r + \lambda_{jR_k}(b_{rR_k,j}-b_{j}^{R}) + \lambda_{jR_k}N_{r,j}^{R} - \kappa_{jR_k}\cdot I_{r,1}^{R} + T + \varepsilon_{r,j}^{R} \\
l_{r,j}^{E} = -\mathbf{u}_{r}^{E}\cdot\mathbf{r} + t_r + \lambda_{jE}(b_{rE,j}-b_{j}^{E}) + \lambda_{jE}N_{r,j}^{E} - \kappa_{jE}\cdot I_{r,1}^{E} + T + \varepsilon_{r,j}^{E} \\
l_{r,j}^{C} = -\mathbf{u}_{r}^{C}\cdot\mathbf{r} + t_r + \lambda_{jC}(b_{rC,j}-b_{j}^{C}) + \lambda_{jC}N_{r,j}^{C} - \kappa_{jC}\cdot I_{r,1}^{C} + T + \varepsilon_{r,j}^{C}
\end{cases}
\tag{1}
$$


$$
\begin{cases}
p_{r,j}^{G} = -\mathbf{u}_{r}^{G}\cdot\mathbf{r} + t_r + c\cdot d_{rG} + \kappa_{jG}\cdot I_{r,1}^{G} + T + e_{r\,j}^{G}, \\
p_{r,j}^{R_k} = -\mathbf{u}_{r}^{R}\cdot\mathbf{r} + t_r + c\cdot d_{rR_k} + \kappa_{jR_k}\cdot I_{r,1}^{R} + T + e_{r\,j}^{R} \\
p_{r,j}^{E} = -\mathbf{u}_{r}^{E}\cdot\mathbf{r} + t_r + c\cdot d_{rE} + \kappa_{jE}\cdot I_{r,1}^{E} + T + e_{r\,j}^{E}, \\
p_{r,j}^{C} = -\mathbf{u}_{r}^{C}\cdot\mathbf{r} + t_r + c\cdot d_{rC} + \kappa_{jC}\cdot I_{r,1}^{C} + T + e_{r\,j}^{C},
\end{cases}
\tag{2}
$$

where $r$ and $j$ refer to receiver and frequency, respectively; The capital indices $G, R, E$, and $C$ refer to
the satellites of GPS, GLONASS, Galileo, and BeiDou, respectively; $R_k$ denotes the GLONASS
satellite with frequency factor $k$; $l_{r,j}$ and $p_{r,j}$ denote the "observed minus computed" phase and
pseudorange observables; $\mathbf{u}_{r}^{s}$ is the unit vector in the receiver to satellite direction; $\mathbf{r}$ denotes the
vector of the receiver position increments relative to the a priori position, which is used for linearization;
$t_r$ is the receiver clock bias; $N_{r,j}$ is the integer ambiguity; $b_j$ are the uncalibrated phase delays; $\lambda_j$ is
the wavelength; the ionospheric delays $I_j$ at different frequencies can be expressed as
$I_j = \kappa_j \cdot I_1, \kappa_j = \lambda_j^2 / \lambda_1^2$; and $T$ is the slant tropospheric delay. Due to the different frequencies and
signal structures of each individual GNSS, the code biases $d_{rG}$, $d_{rR_k}$, $d_{rE}$, and $d_{rC}$ are different for
each multi-GNSS receiver. These inter-system biases (ISB) and inter-frequency biases (IFB) of the
GLONASS satellites with different frequency factors have to be estimated or corrected for a combined
processing of multi-GNSS observations. $e_{r,j}$ and $\varepsilon_{r,j}$ denote the sum of measurement noise and





multipath effects of pseudorange and phase observations, respectively. The phase center offsets and
variations, the tidal loading, and the phase wind-up are corrected with the models according to Kouba

(2009).

The slant total delay $T$ can be described as the sum of the hydrostatic and non-hydrostatic/wet

components, and the horizontal gradient components (Chen and Herring, 1997),

$$T = mf_h \cdot ZHD + mf_{nh} \cdot ZWD + mf_G \cdot (G_{ns} \cdot \cos(a) + G_{ew} \cdot \sin(a)) \qquad (3)$$

where ZHD and ZWD denote the zenith hydrostatic and non-hydrostatic/wet delays, respectively, $mf_h$
and $mf_w$ are the hydrostatic and non-hydrostatic mapping functions (here Global Mapping Functions
(GMF), Boehm et al., 2006), $mf_G$ represents the gradient mapping function, $G_{ns}$ and $G_{ew}$ are the
north-south (NS) and east-west (EW) delay gradients, respectively, and $a$ is the azimuth of the line of
sight of the individual observation.

Concerning the approach for tropospheric delay modeling, two PPP scenarios are applied in this

study: one is the standard PPP processing with tropospheric delays estimated as unknown parameters,
and the other is the developed NWM-constrained PPP algorithm which utilizes tropospheric delay
parameters derived from ECMWF. For the standard PPP processing, a priori ZHD is calculated by use
of the empirical models (Saastamoinen, 1973) based on the provided meteorological information (here
Global Pressure and Temperature 2 model (GPT2), Lagler et al., 2013) at a given location. Owing to the
high variability of the water vapor distribution, the ZWD is estimated as an unknown parameter in the
adjustment together with the other parameters, such as the station coordinates. The horizontal



tropospheric gradients, $G_{ns}$ and $G_{ew}$, are also estimated, both with a temporal resolution of 24 hours.
The parameters estimated in the standard PPP processing include station coordinates, ambiguity
parameters, receiver clock corrections, ZWD, and gradient components, all of which are adjusted in a
sequential least squares filter. For the standard multi-GNSS PPP processing, the parameter vector $\mathbf{X}$
can be described as,
$$\mathbf{X} = \begin{pmatrix} \mathbf{r} & t_r & ZWD & G_{ns} & G_{ew} & d_{rE} & d_{rC} & d_{rR_k} & \mathbf{I}_{r,1}^s & \mathbf{N}_{r,j}^s \end{pmatrix}^T \qquad (4)$$

For the NWM-constrained PPP approach, ZHD, hydrostatic and non-hydrostatic mapping functions
are derived from the ECMWF analysis. The ZWD from ECMWF is considered as the a priori value for
the wet delays, while a residual wet delay is estimated during the parameter estimation process in order
to account for possible imperfections inherent in the NWM. The horizontal gradients are also derived
from the ECMWF analysis and are fixed during the processing. In this approach, the unknown
parameters are station coordinates, ambiguity parameters, receiver clock corrections, and the residual
ZWD error. The latter is modeled as a random walk process with a priori constraints related to the
accuracy of tropospheric delay parameters derived from ECMWF. Accordingly, the parameter vector
$\mathbf{X}$ in the NWM-constrained multi-GNSS PPP can be expressed as,
$$\mathbf{X} = \begin{pmatrix} \mathbf{r} & t_r & Resi_{ZWD} & d_{rE} & d_{rC} & d_{rR_k} & \mathbf{I}_{r,1}^s & \mathbf{N}_{r,j}^s \end{pmatrix}^T, \quad Resi_{ZWD} \sim N(0, \sigma_{ZWD}^2) \qquad (5)$$

where $Resi_{ZWD}$ denotes the residual ZWD error, and $\sigma_{ZWD}^2$ is the variance of the *ZWD*.
In order to carry out a rigorous multi-GNSS analysis including the estimation of the inter-system
and inter-frequency biases, the observables from the four individual GNSS are processed together in a



single weighted least squares estimator. For the two multi-GNSS PPP scenarios, the receiver position
increment **r** is estimated as static parameter on a daily basis. The receiver clock bias $t_r$ is estimated
as white noise, and the inter-system and inter-frequency code biases are estimated as parameters on a
daily basis. The ZWD or the residual wet delay $Resi_{ZWD}$ are modeled as piece-wise constant
parameters (with a temporal resolution of two hours). The code biases for GPS satellites are set to zero
to eliminate the singularity between receiver clock and code bias parameters. All the estimated biases of
the other systems are relative to those of the GPS satellites. The phase ambiguity parameters $\mathbf{N}_{r,j}^{s}$, which
absorb the phase delays $b_j$, are estimated as constants for each continuous arc. With the combination
of the dual-frequency raw phase and pseudorange observations, the ionospheric delays $\mathbf{I}_{r,1}^{s}$ are
considered as estimated parameters for each satellite-site pair and each epoch. Besides, an
elevation-dependent weighting and a cut-off elevation angle of 5 °are applied.

**3 Results and analysis**
**3.1 Comparison between ECMWF and IGS ZTD**
In this section, the quality of tropospheric zenith delay parameters derived from ECMWF analysis is
evaluated by comparing with the zenith path delay products offered by IGS. Specifically, the ECMWF
ZTD for 34 globally-distributed stations from the IGS MGEX network during September 2015 are
validated by the official IGS ZTD products which are provided with a temporal resolution of five
minutes. As the ECMWF ZTD are sampled every six hours, we do not interpolate in time but restrict the





comparison to the ECMWF data epochs.
As typical examples, the ZTD series derived from ECMWF and IGS at stations KIRU (Kiruna,
Sweden) and NNOR (New Norcia, Australia) are shown in Figure 2. The ECMWF ZTD are represented
through black triangles, while the IGS ZTD are displayed by red squares. One can notice that the
ECMWF ZTD show good agreement with the IGS ZTD in general. Most of the peaks in the ZTD series,
which are mainly caused by rapid changes of the water vapor content above a station, are captured by
ECMWF and IGS solutions.
The corresponding linear correlations between the ECMWF and the IGS ZTD at stations KIRU and
NNOR are illustrated in Figure 3. It can be seen that ZTD from the two solutions are highly correlated,
with the correlation coefficients being about 0.93 and 0.97, respectively. Figure 4 presents the
distribution of ZTD differences between ECMWF and IGS for the two stations during the same period.
One can notice that the ZTD differences mainly range from -15 to 15 mm for station KIRU, and vary
between -10 and 10 mm for station NNOR. The mean biases of the ZTD differences between the two
solutions are -3.52 and 3.31 mm for the two stations and the root-mean-square (RMS) values of the
ZTD differences are 8.68 and 6.39 mm, respectively, showing an agreement at the mm-level.
Figure 5 illustrates the map of station specific mean biases and RMS values of ZTD differences
between ECMWF and IGS for all stations. One can notice that the mean biases are within ±15 mm,
and that a better agreement between the ECMWF and IGS ZTD for the high-latitude stations than for
the low-latitude stations can be observed. The RMS values of the ZTD differences are less than 22 mm,
indicating a good agreement between the two solutions. Likewise, the RMS values present a significant
latitude dependence, which is smaller for high-latitude stations and larger for low-latitude stations,
resulting from the distribution of atmospheric water vapor content with respect to the stations' latitudes.
The RMS values for stations in high-latitude regions are generally below 15 mm, while the ones for the
stations in low-latitude regions can reach up to 22 mm. For an enhanced perspective, the RMS values of
ZWD differences between ECMWF and IGS are shown as a function of the geographical latitudes in
Figure 6, where a fitted parabola is also displayed in black. It can be clearly seen that the RMS values
reveal strong dependence on geographical latitudes, which are larger in low-latitude (moist) regions and
smaller in high-latitude (dry) regions.

## 3.2 Multi-GNSS PPP results


To investigate the performance of applying tropospheric delay parameters derived from ECMWF into
multi-GNSS PPP, two PPP scenarios including the standard PPP and the NWM-constrained PPP are
carried out for comparing and validating, following the data processing algorithms presented in Sect.
2.3. Observational data from stations of the IGS MGEX network (see Fig.1) in September 2015 are
considered in this study.
As an example, Figure 7 illustrates the estimated north/east/up coordinates obtained from the two
multi-GNSS PPP processing method at station WIND (Windhoek, Namibia, 22.57 ° S, 17.09 ° E) on
September 12, 2015. As a reference, positioning results derived from the stand-alone GPS PPP are also
displayed applying similar strategies as the multi-GNSS processing. The standard PPP solutions are





shown by black triangles, while the NWM-constrained PPP solutions are shown by red squares. The left
figures show the multi-GNSS results. One can notice that it takes about 17 min for the
NWM-constrained multi-GNSS PPP to achieve an accuracy of a few centimeters for the north
component, in comparison to 25 min in case of the standard PPP solution. The convergence time is
shortened by about 32.0 % by using the NWM-derived tropospheric delay parameters. Meanwhile, the
positioning series of the standard PPP solution show a larger jump than that of the NWM-constrained
PPP solution before the convergence. As for the east component, centimeter-level accuracy is
achievable with a convergence time of about 40 min for the standard vs. 25 min for the
NWM-constrained PPP solution. Accordingly, the solution is improved in terms of convergence time by
about 37.5 % with the NWM-constrained PPP. For the vertical component, it can be seen that the
convergence time is also clearly reduced by applying the NWM-constrained PPP. A convergence time of
about 20 min and 15 min is required to reach decimeter-level accuracy for the standard PPP solution and
the NWM-constrained PPP solution, respectively, indicating an improvement of about 25.0 % when
applying the NWM-constrained PPP. In addition, the positioning series exhibit much more jumps and
fluctuations with the standard PPP solution, in particular before the solution convergence, which get
significantly improved when the NWM-constrained PPP is performed.

As shown in the right figures, the positioning performance, not only the convergence time but also

the positioning series, gets remarkably improved with the multi-GNSS processing (left figures)
compared to the GPS-only solution. For the standard GPS PPP, an accuracy at the centimeter-level is
obtainable with a convergence time of about 50 min and 60 min for the north and east components,
respectively. In comparison, the convergence time is improved by about 60 % and 33.3 %, when the





standard multi-GNSS PPP (about 20 min and 40 min for north and east components, respectively) are
carried out. Meanwhile, it takes about 20 min and 40 min for the NWM-constrained GPS PPP solution
to reach a comparable centimeter-level accuracy for the north and east components, respectively,
shortening the solution convergence time to about the same extent as the multi-GNSS combination. In
the standard GPS PPP solution, a convergence time of about 50 min is required for the vertical
component to achieve an accuracy of a few decimeters, in comparison to 20 min in case of the standard
multi-GNSS PPP solution. The convergence time is reduced by about 60 % attributing to the
multi-GNSS fusion. The convergence time for the NWM-constrained GPS PPP solution is about 10 min
for the vertical component, revealing an improvement of up to 80 % compared to the standard GPS PPP
solution. In addition, it can be found that the NWM-constrained PPP reveals significant contribution to
improving the positioning series of all three components, showing more stable and less fluctuated
results.

In Figure 8, the statistical results of the multi-GNSS PPP solutions are presented with different

session lengths (5, 8, 10, 15, 17, 20, 25, 30, 40, 50, and 60 min). The RMS values of the positioning
results for the north/east/up components are calculated for selected stations from the MGEX network
over a sample period from September 1 to September 30, 2015. The standard PPP solution is shown in
orange, the NWM-constrained PPP solution in olive. Obviously, the positioning accuracy of each
component improves along with the increase of the session length for both PPP scenarios. In general,
the positioning accuracy of the north component is better than that of the east and the vertical
components, while the vertical component performs the worst, which may be attributed to the



configuration of the satellite constellation.
For the north component, the RMS values obtained from the NWM-constrained PPP solution are
smaller than the ones from the standard PPP solution at the same session length, especially before
convergence. The positioning accuracy achieved with the NWM-constrained PPP is improved by about
2.5 % compared to the one with the standard PPP. Besides, a convergence time of about 20 min and 25
min is observed for the NWM-constrained PPP solution and the standard PPP solution, respectively: an
improvement of about 20.0 %. In terms of the east component, higher accuracy can be found again for
the NWM-constrained PPP solution, with the RMS values reduced by about 12.1 %. Meanwhile, the
standard PPP solution takes about 25 min to achieve an accuracy of a few centimeters; the same level is
reached in about 17 min for the NWM-constrained PPP solution, a significant reduction in the
convergence time of about 32.0 %.
As for the up component, it can be noticed that the positioning accuracy achieved from the
NWM-constrained PPP solution is obviously higher than that from the standard PPP solution, an
improvement of about 18.7 %. More than 20 min are required for the standard PPP solution to reach an
accuracy of a few decimeters, while the NWM-constrained PPP solution achieves the same accuracy in
less than 15 min, indicating an improvement of more than 25 %.





## 4 Conclusions

We developed a NWM-constrained PPP processing system where tropospheric delay parameters derived
from the ECMWF analysis were applied to multi-GNSS precise positioning. Observations of stations
from the IGS MGEX network were processed, with both standard PPP and the developed
NWM-constrained PPP algorithm. The accuracy of the tropospheric delays derived from ECMWF was
assessed through comparisons with the IGS final tropospheric delay products at all IGS MGEX stations.
The positioning performance, including convergence time and positioning accuracy, achieved with the
NWM-constrained PPP were investigated. The benefits of applying tropospheric delay parameters from
the NWM to improve multi-GNSS PPP were demonstrated by comparing with the standard PPP
solution.
Our results show that the mean biases between the ECMWF and IGS ZTD are within $\pm15$ mm,
while the RMS values of the ZTD differences are less than 22 mm, indicating a good agreement
between the two solutions. Besides, a better agreement for the high-latitude stations than for the
low-latitude stations can be noticed. Both the mean biases and RMS values are smaller for high-latitude
(dry) regions and larger for low-latitude (moist) regions, revealing significant latitude dependence.
These may be accounted for by the distribution of atmospheric water vapor with respect to station
latitudes. Furthermore, most of the peaks in the ZTD series, which are attributed to the rapid changes of
the water vapor content above a given station, can be captured by both ECMWF and IGS solutions.
For the north component, it takes about17 min for the NWM-constrained multi-GNSS PPP to





achieve an accuracy of a few centimeters, in comparison to 25 min for the standard PPP solution,
showing an reduction of the convergence time of about 32.0 %. The centimeter-level accuracy is
achieved for the east component after a convergence time of about 40 min and 25 min from the standard
and the NWM-constrained PPP solutions, respectively. The convergence time is shortened by 37.5 %
with the NWM-constrained PPP. For the vertical component, a convergence time of about 20 min and
15 min is required to reach decimeter-level accuracy for the standard PPP solution and the
NWM-constrained PPP solution, respectively, indicating an improvement of about 25.0 % when
applying the NWM-constrained PPP. Meanwhile, the positioning series get significantly improved with
the NWM-constrained PPP solution, displaying less jumps and fluctuations, especially before the
solution convergence and for the vertical component.

Besides, the positioning performance of the multi-GNSS processing achieves remarkable

improvement compared to the GPS-only solution. In comparison with the standard GPS PPP, the
convergence time is improved by about 60 %, 33.3 %, and 60 % for the north, east, and vertical
components, respectively, when conducting the standard multi-GNSS PPP. Meanwhile, when the
NWM-derived tropospheric delay parameters are implemented instead of the standard GPS PPP, the
convergence time gets shortened to the same extent as the multi-GNSS processing (by about 60 % and
33.3 %) for the north and east components. An improvement of convergence time up to 80 % for the
vertical component can also be observed. In addition, the NWM-constrained GPS PPP shows significant
contribution to improving the positioning series for all three components, with much more stable and

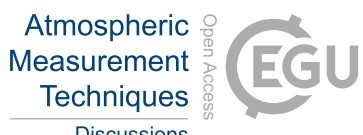

less fluctuated results in particular for the vertical component. According to the results, it can be
concluded that the performance of precise positioning benefits greatly from the multi-GNSS fusion in
comparison to the stand-alone GPS solution, which can be further improved when the tropospheric
delay parameters derived from NWM are implemented to the multi-GNSS PPP processing.
Furthermore, the positioning accuracy obtained from the NWM-constrained multi-GNSS PPP
solution is also improved in comparison with the standard PPP solution with the same session length, in
particular before convergence. After the convergence of the solution, an improvement of positioning
accuracy resulting from the NWM-constrained PPP solution of about 2.5 %, 12.1 %, and 18.7 % for the
north, east, and vertical components, respectively, can be found.
In future studies, we will investigate the performance of applying tropospheric delay parameters
derived from the NWM into precise positioning with other single satellite navigation systems, such as
the Russian GLObal NAvigation Satellite System (GLONASS) and the Chinese BeiDou Navigation
Satellite System (BDS). Another research focus is the evaluation of the accuracy and performance of
different numerical weather models, in order to find the most appropriate one to improve precise GNSS
positioning.

**Acknowledgements.** Many thanks go to the International GNSS Service (IGS) for providing
multi-GNSS data and the IGS final tropospheric products. The ECMWF data are provided to GFZ via
the German Weather Service (DWD).





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



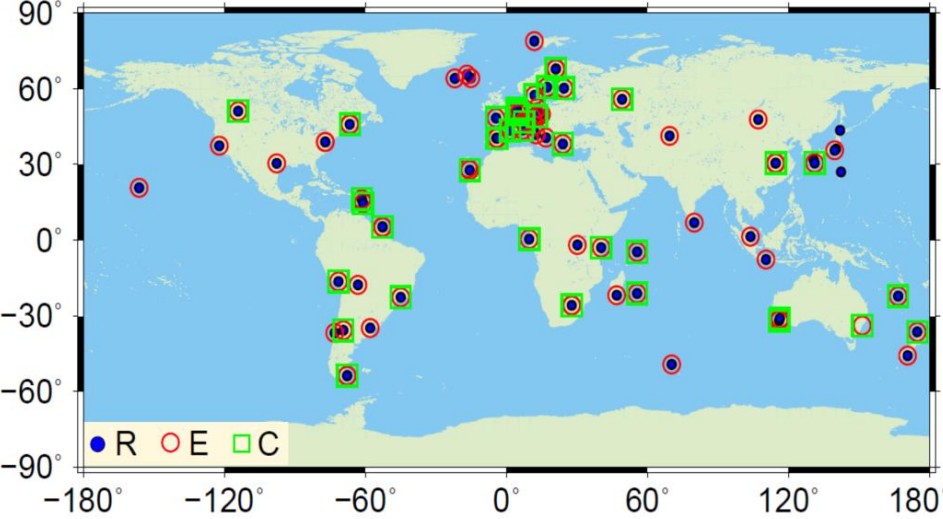


Figure 1: The geographical distribution of the MGEX stations and their supported navigation satellite constellations. The symbols "R", "E", and "C" refer to GLONASS, Galileo, and BeiDou, respectively, while GPS can be tracked by each station.










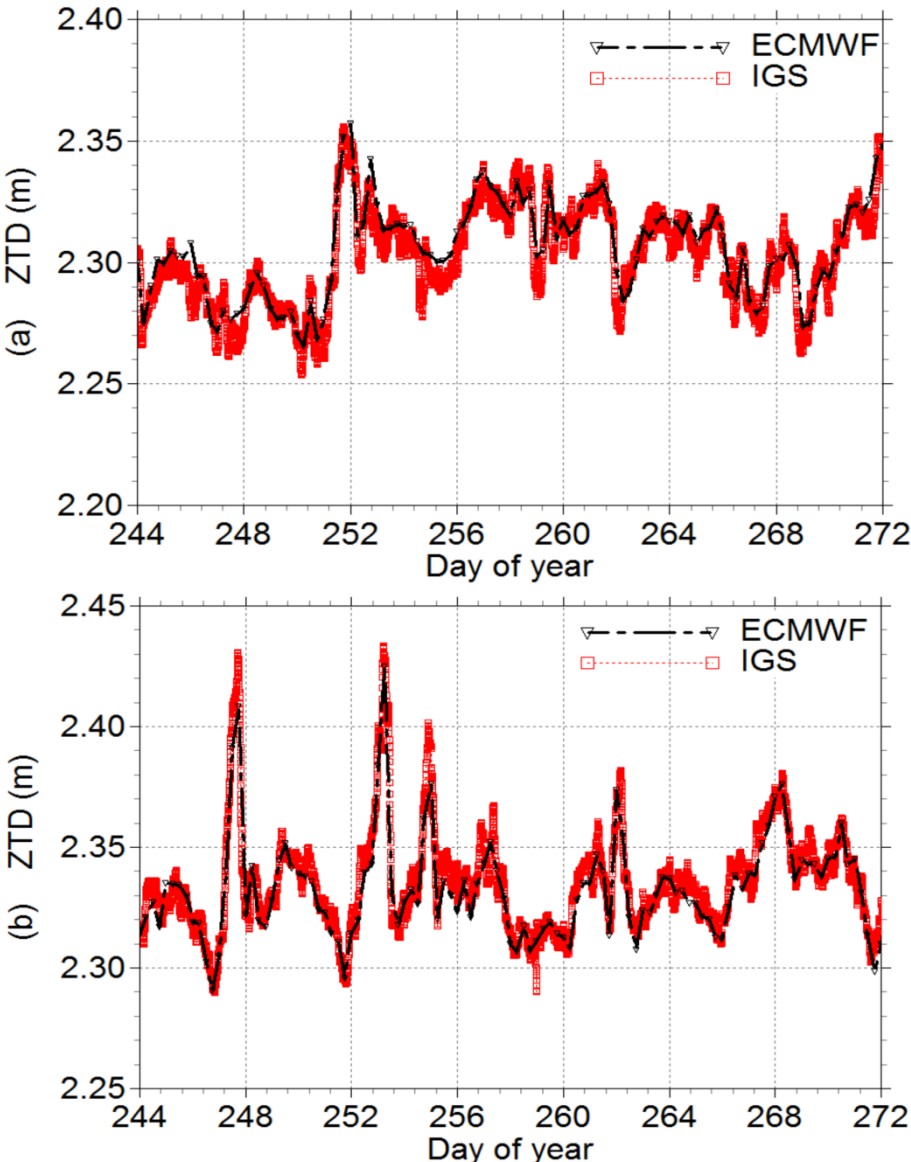



Figure 2. The time series of ECMWF and IGS ZTD at stations KIRU (a) and NNOR (b) for September
(day of year (DOY) 244-272) 2015. The ECMWF ZTD are shown by black triangles, while the IGS
ZTD are displayed by red squares.





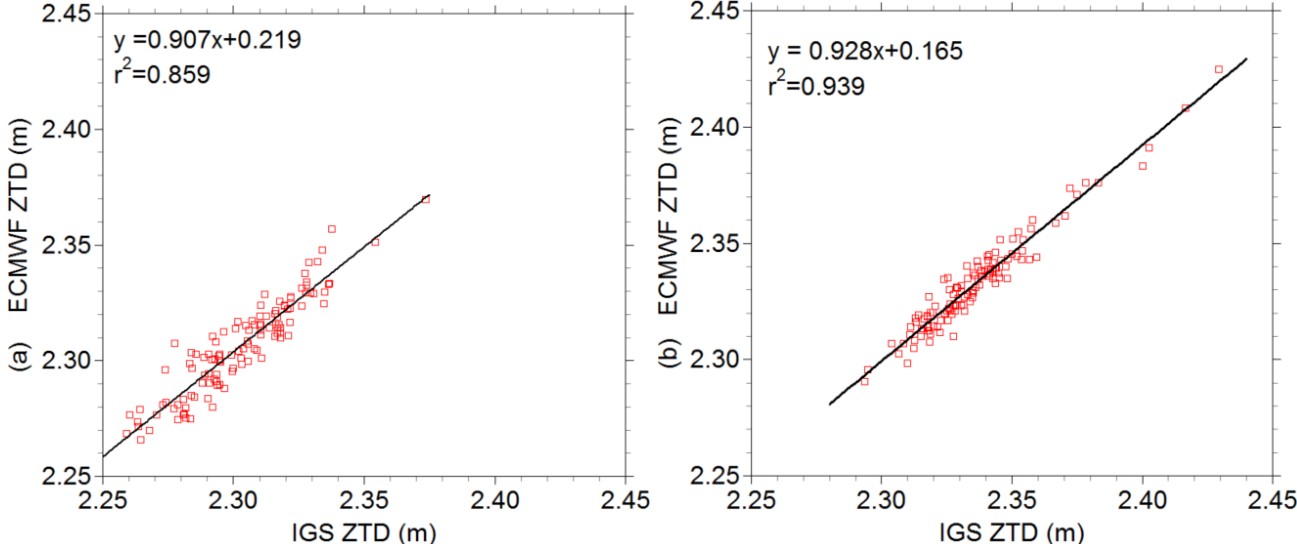

Figure 3. Scattergram of ECMWF and IGS ZTD at stations KIRU (a) and NNOR (b). The vertical and horizontal axes show ECMWF and IGS ZTD (m), respectively. The correlation coefficients (r) and the results of a linear regression are also displayed.





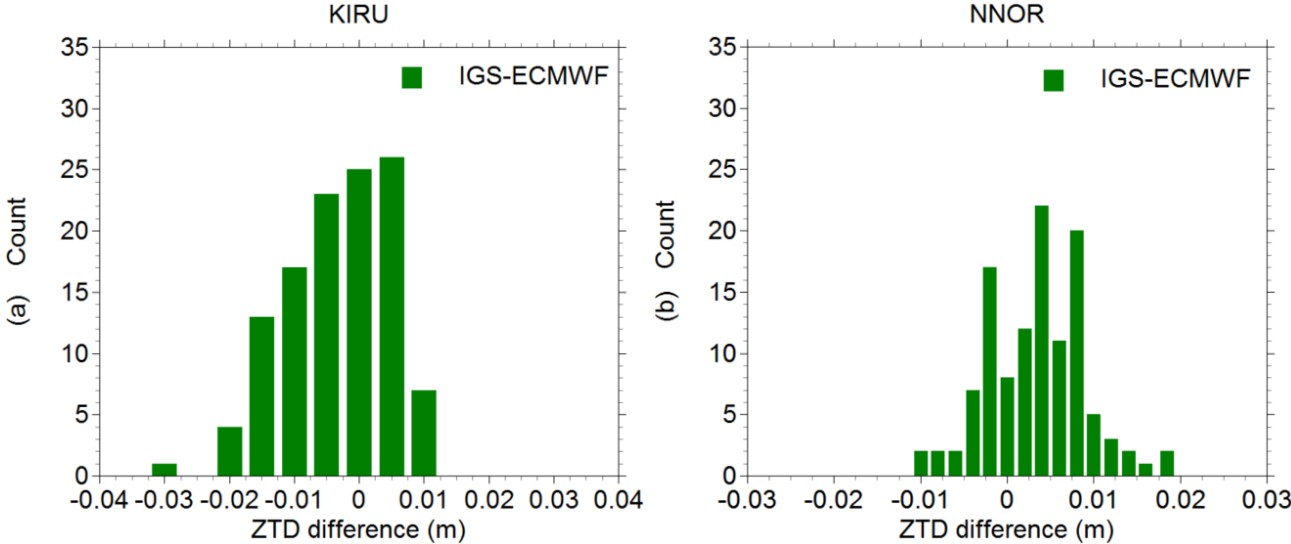


Figure 4. Distribution of ZTD differences between ECMWF and IGS ZTD at stations KIRU (a) and

NNOR (b) for DOY 244-272, 2015.




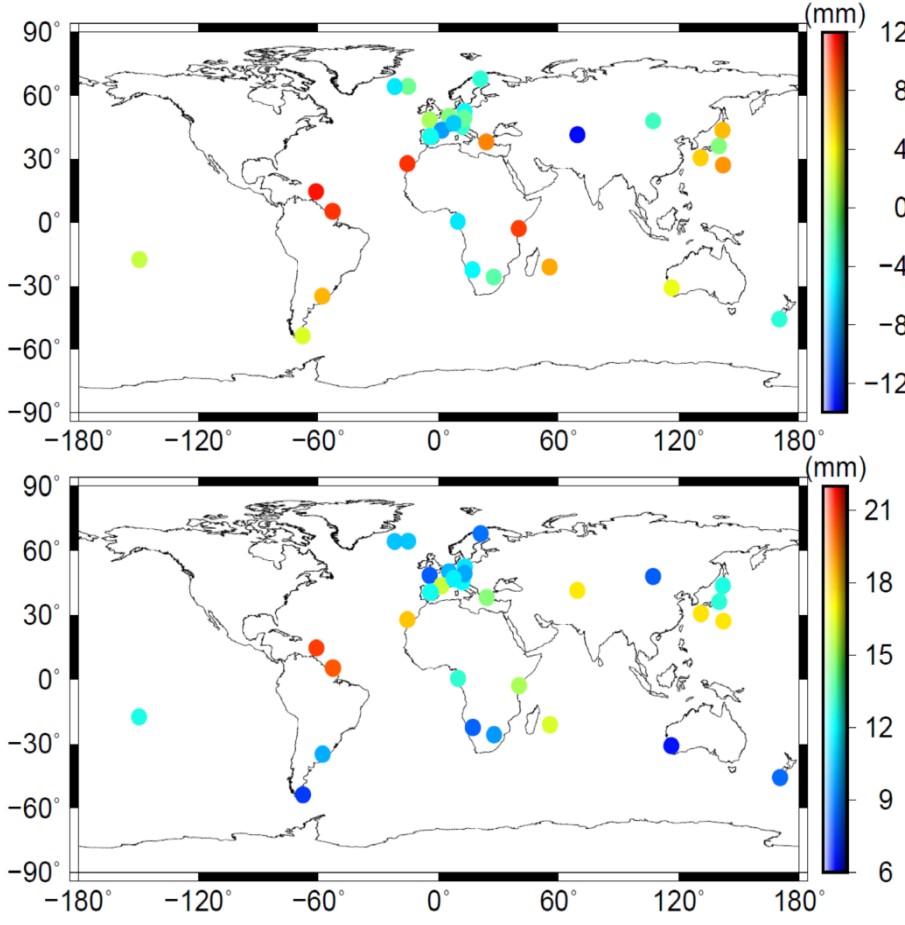


Figure 5. The map of the station-specific mean biases (top) and RMS values (bottom) of ZTD differences between ECMWF and IGS for DOY 244-272, 2015.






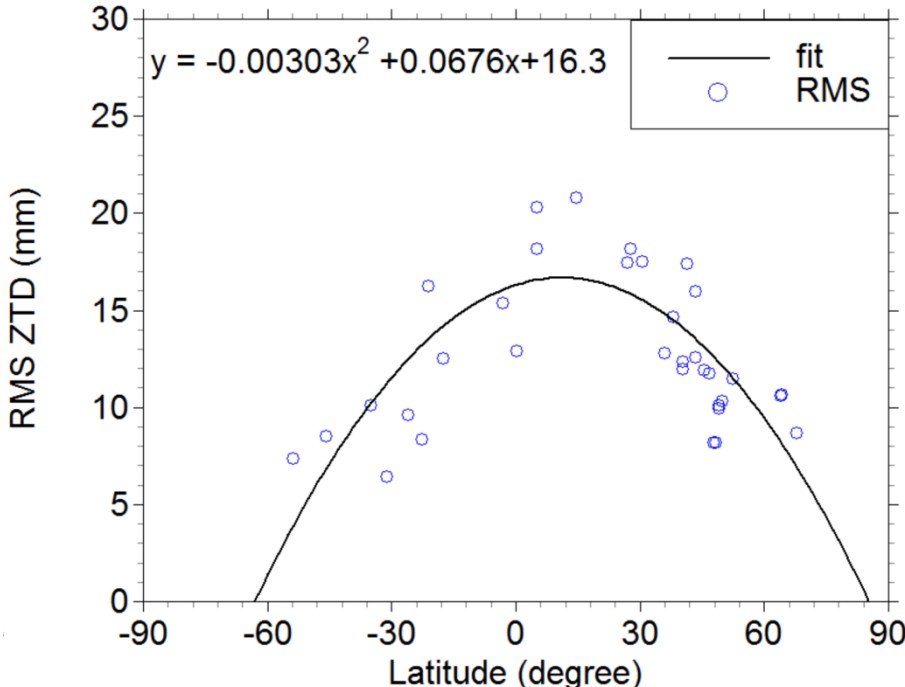


Figure 6. The RMS values of ZTD differences between ECMWF and IGS as a function of geographical

latitudes. A fitted second-order polynomial is also shown in black.









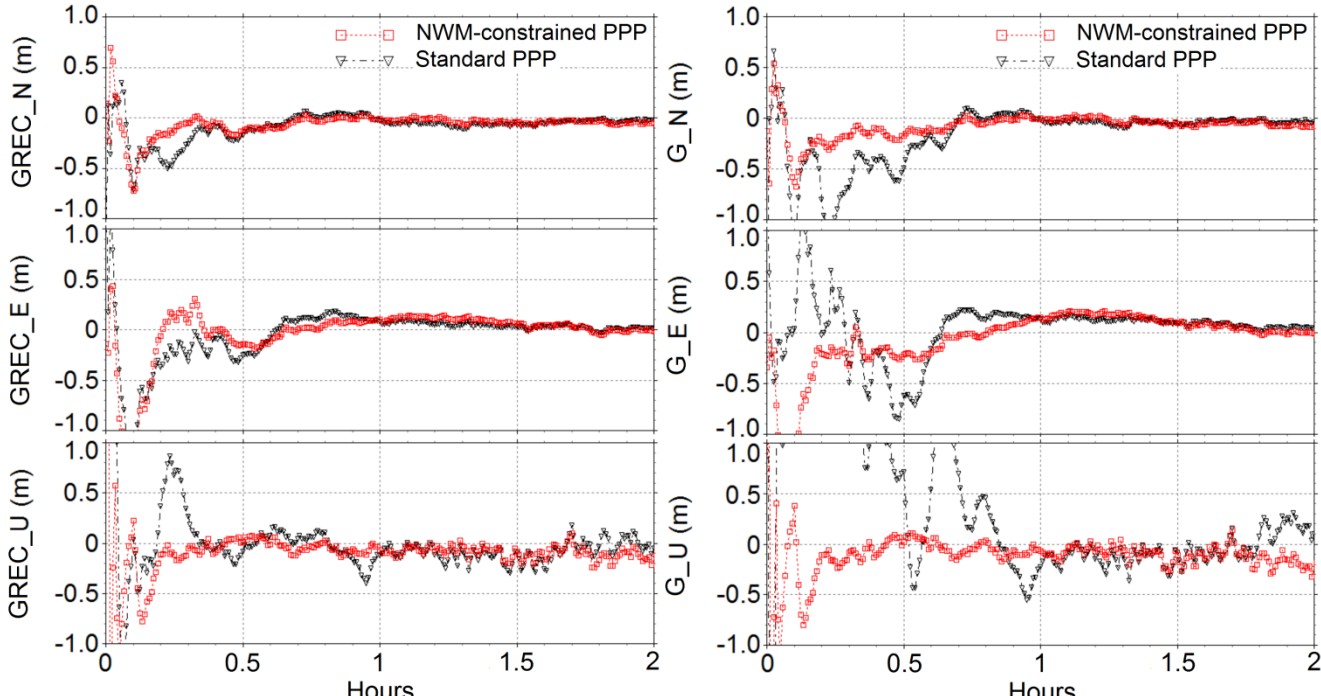

Figure 7. The multi-GNSS PPP ("GREC") solution (left) and the stand-alone GPS PPP ("G") solution (right) at station WIND (Windhoek, Namibia, 22.57 °S, 17.09 °E) on September 12, 2015 (DOY 255 of 2015). The standard PPP solutions are shown by black triangles, while the NWM-constrained PPP solutions are shown by red squares.





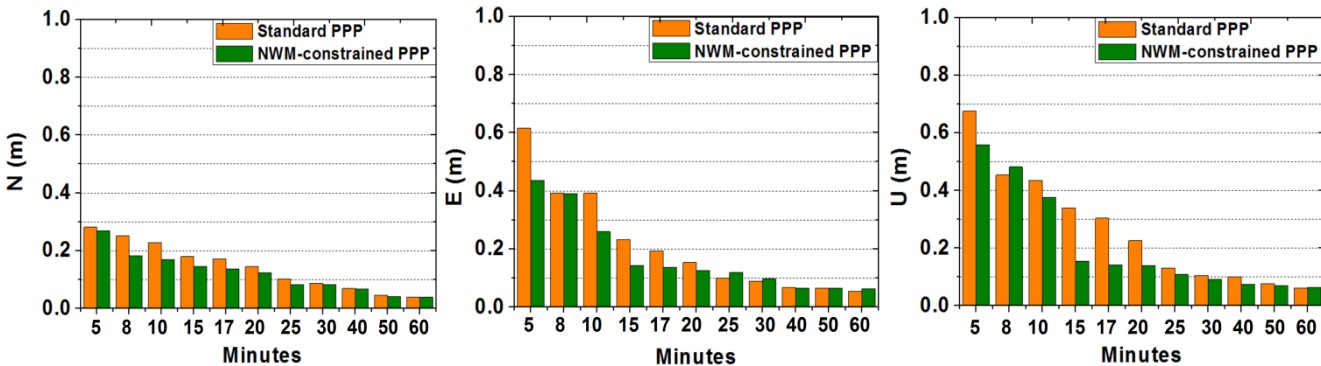

501

Figure 8. The RMS values for the north/east/up components with multi-GNSS PPP solution, showing at different session lengths (5, 8, 10, 15, 17, 20, 25, 30, 40, 50, and 60 min) for selected MGEX stations from September 1 to September 30, 2015. The standard PPP solution is shown in orange, the NWM-constrained PPP solution in olive.