# Peer review of "Tropospheric delay parameters from numerical weather models"

_Atmospheric Measurement Techniques, 2016_

## Referee Comment (RC1) · Anonymous Referee #1 · 28 Jun 2016

The comment was uploaded in the form of a supplement:
http://www.atmos-meas-tech-discuss.net/amt-2016-176/amt-2016-176-RC1-supplement.pdf

---

## Referee Comment (RC2) · Anonymous Referee #2 · 19 Jul 2016

**General comments:**

The manuscript describes novel method of exploiting external tropospheric corrections in standalone and multi-GNSS Precise Point Positioning and assesses the impact on position accuracy and the convergence time. The method is clearly described, results show positive impact of both multi-GNSS and NWM-constrained PPP compared to standard and standalone PPP which is clearly demonstrated in presented figures and described statistics (how the statistics were calculated and derived remained however unclear). The manuscript is well organized and understandable; however, Section 3.2 summarizing the results is difficult to follow. I would suggest completing/revising the manuscript with some more detail/clear descriptions in three directions:

- More details about PPP processing/modelling in Section 2.3 (see also Specific comments: constraining values, multi-GNSS observation weighting, random walk vs. piece-wise constant modelling, elevation-dependent weighting model etc, precise products and PPP software used).

- Specification of the methodology used for assessing the position accuracy and the convergence time (see also Specific comments: the reference position, criteria for the assessing of accuracy at 'decimetre/centimetre-level' and criteria for the estimating of convergence time, included/excluded stations or results in statistics). E.g. could be described at the beginning of Section 3.

- Extremely long part (~4 pages, Section 3.2 and Conclusion) concerns of the description on the assessment of convergence time and position accuracy which I found very difficult to following thoughtfully. I suggest shortening the part avoiding very similar phrasing (a reader gets easily lost here) while preferably adding more concise overview (e.g. table) about resulting statistics for the performance of standalone & multi-GNSS and standard & NWM-constrained PPP. In my opinion, it should significantly facilitate understanding presenting results and their representativeness.

**Specific comments and technical corrections:**

Page 2, line 33: Suggested to use 'PPP method' instead of 'PPP technique'.

Page 4, line 64: GPS + GLONASS + … (space between GPS+)

Page 8, line 141: on 137 vertical model levels – is this correct? According to the documentation the ERA-Interim has 60 model (e.g. ERA Report series, The ERA-Interim archive Version 2).

Page 9, line 154: respectively; the capital …. (start with lower case)

Page 9,eq.1: missing explanation for $b_r$

Page 9, lines 160-161: use $I_{r,j}$ instead of $I_j$ .

Page 11: Add information about how NWM parameters were interpolated in time.

Page 11: What value is used as the ZWD variance for constraining $ZWD_{resi}$?

Page 11, line 197: Here it is written that $ZWD_{resi}$ is modeled as a random walk process, later (Page 12, line 207) it is mentioned that $ZWD_{resi}$ is modeled as piece-wise linear function. Please clarify.

Page 12, line 208: Please, comment use of 2-hour piece-wise constant in ZWD modeling since it is rather long interval for using constant value for modeling the troposphere and may cause the degradation in performance.

Page 12: What was a relative weighting (if any) applied for phase and code observations for different systems?

Page 12: Which elevation-dependent weighting function was used?

Page 12, line 219: add the reference to IGS tropo products (Byram et al., ION, 2011).

Page 13, line 227: ECMWF ZTDs show good agreement with the IGS ZTDs (plural otherwise shows)

Page 13, lines 245-250: A strong latitudinal dependency of ZTDs from the MetOffice UK global model with respect to GPS ZTDs was described in Dousa J and Bennitt G, GPS solutions, 2012.

Pages 15-16 + Fig 7: How were percentages 32%, 37.5%, 25% derived – supposed the values are calculated as representative over all the stations, but currently it seems they are reported for the station WIND (figure). Anyways, please provide more details about the calculation of statistics (i.e. underlying their representativeness) in terms of number of data/stations used (included/excluded), criteria applied for achieving the convergence, 'centimetre/decimetre-level' accuracy etc.

Page 17, line 306: … before the convergence. (the article)

Page 17, line 305: … position accuracy … It should be also explained which reference values were used in the accuracy assessment. Here, I suppose the 'position accuracy' should mean 'North component accuracy' since East & Up follows and no more North.

Page 18, line 340: about:17 (add space)

Page 19, line 366: … before the convergence (the article)

Page 20, line 367: … 2.5%, 12.1% and 18.7% - only here it is mentioned the values are gained 'after' the reaching convergence (line 366). How the convergence time is determined? Neither abstract nor Section 3.2 mentioned it concerns the period after reaching the convergence.

---

## Author Comment (AC1) · 11 Oct 2016

**Manuscript:**

Atmos. Meas. Tech. Discuss., doi:10.5194/amt-2016-176, 2016 Lu, Tropospheric delay parameters from numerical weather models for multi-GNSS precise positioning

Reviewer #1:

**General Comment:**

The manuscript covers a study of demonstrating the improvement of the GNSS PPP solutions over 'standard PPP' approach in terms of reduced convergence time, overall precision, and overall reliability by taking advantage of external available tropospheric delay parameters. Though approach has been used by analysis center like SOPAC for many year for GPS only realtime epoch by epoch solutions (never formally published), the current study reported all necessary details of the theory together with the statistics of case studies in a multi-GNSS setting which is extremely challenging in its implementation. The description of the method with its reasoning of the logic behind is sound. The results and derived statistics are convincing. The overall presentation including the drown conclusions are very clear.
Answer: Many thanks for your comments.

**Specific Comments (P denotes page L denotes Line):**

P2L32: Comment: the quotations in () appear unnecessary as there are many related contributions available. Single out these two sounds somewhat self-promoting.
Answer: Removed.

P9L152: Comment: the $\mathbf{b}$r in eq (1)are not verbally defined in the following text.

Answer: "$b_{r,j}$ and $b_j^s$ are the uncalibrated phase delays for receivers and satellites, respectively" was added

P11L185: Comment: It is unlikely that the tropospheric gradients will remain constant over 24 hour period. More realistic tropospheric gradient settings would be 4 times or at least 2 times a day. Of course the impact of using 24 hour tropospheric gradient parameters would be negligible for this study。
Answer: Thanks a lot for your comments.

---

## Author Comment (AC2) · 11 Oct 2016

**Manuscript:**

Atmos. Meas. Tech. Discuss., doi:10.5194/amt-2016-176, 2016 Lu, Tropospheric delay parameters from numerical weather models for multi-GNSS precise positioning

**General comments:**

The manuscript describes novel method of exploiting external tropospheric corrections in standalone and multi-GNSS Precise Point Positioning and assesses the impact on position accuracy and the convergence time. The method is clearly described, results show positive impact of both multi-GNSS and NWM-constrained PPP compared to standard and standalone PPP which is clearly demonstrated in presented figures and described statistics (how the statistics were calculated and derived remained however unclear). The manuscript is well organized and understandable; however, Section 3.2 summarizing the results is difficult to follow. I would suggest completing/revising the manuscript with some more detail/clear descriptions in three directions:

1) More details about PPP processing/modelling in Section 2.3 (see also Specific comments: constraining values, multi-GNSS observation weighting, random walk vs. piece-wise constant modelling, elevation dependent weighting model etc, precise products and PPP software used).

2) Specification of the methodology used for assessing the position accuracy and the convergence time (see also Specific comments: the reference position, criteria for the assessing of accuracy at 'decimetre/centimetre-level' and criteria for the estimating of convergence time, included/excluded stations or results in statistics). E.g. could be described at the beginning of Section 3.

3) Extremely long part (~4 pages, Section 3.2 and Conclusion) concerns of the description on the assessment of convergence time and position accuracy which I found very difficult to following thoughtfully. I suggest shortening the part avoiding very similar phrasing (a reader gets easily lost here) while preferably adding more concise overview (e.g. table) about resulting statistics for the performance of stand-alone & multi-GNSS and standard & NWM-constrained PPP. In my opinion, it should significantly facilitate understanding presenting results and their representativeness.

Answer: Many thanks for your valuable comments.

1) More details concerning the PPP processing details/modeling were added in the revised manuscript, referring to the answers given in the "Specific comments and technical corrections". "The EPOS-RT software (Ge et al. 2012; Li et al. 2013) is utilized for the GNSS data processing in this study, and the GFZ precise products are used." was also added for illustrating the applied PPP software and precise products.

2) The description: "The post-processing weekly solution is used as the reference position. The convergence time was defined as the time required for the horizontal components to be better than 10 cm, and the one needed for the vertical

component to be better than 20 cm." was added at the beginning of Section 3.2 for the description of method for assessing the positioning accuracy, the reference position, criteria for estimating the convergence time. The stations that are given as an example, as well as included for presenting the averaged statistics are also clarified, such as "The percentages 32%, 37.5%, 25% are obtained for station WIND, while the averaged results from all four-system stations (shown in Figure 1) are displayed in Figure 8."

3) The part of illustrating the GPS and multi-GNSS PPP results, from both the standard & NWM-constrained PPP solution, was shortened and rewritten to a concise description, for facilitating the reading and understanding.

**Specific comments and technical corrections:**

Page 2, line 33: Suggested to use 'PPP method' instead of 'PPP technique'.
Answer: Corrected.

Page 4, line 64: GPS + GLONASS + … (space between GPS+)
Answer: Corrected.

Page 8, line 141: on 137 vertical model levels – is this correct? According to the documentation the ERA-Interim has 60 model (e.g. ERA Report series, The ERA-Interim archive Version 2).
Answer: We used the operational analysis of the ECMWF and not the ERA-Interim. Therefore, the number of model levels is correct.

Page 9, line 154: respectively; the capital …. (start with lower case)
Answer: Corrected.

Page 9, eq.1: missing explanation for br

Answer: "$b_{r,j}$ and $b_j^s$ are the uncalibrated phase delays for receivers and satellites, respectively" was added.

Page 9, lines 160-161: use Ir,j instead of Ij .
Answer: Corrected.

Page 11: Add information about how NWM parameters were interpolated in time.
Answer: We added: "These ECMWF-derived tropospheric delay parameters are linearly interpolated to be applied in the GNSS processing."

Page 11: What value is used as the ZWD variance for constraining ZWD_resi?
Answer: As we described in the manuscript: "In this approach, the unknown parameters are station coordinates, ambiguity parameters, receiver clock corrections, and the residual ZWD. The latter is modeled as a random walk process with a priori

constraints related to the accuracy of tropospheric delay parameters derived from ECMWF", the constrains of the residual ZWD is referred to the accuracy of ECMWF-derived parameters with respect to the IGS tropospheric products, which is a function of station latitudes as illustrated in Figure 6.

Page 11, line 197: Here it is written that ZWDresi is modeled as a random walk process, later (Page 12, line 207) it is mentioned that ZWDresi is modeled as piece-wise linear function. Please clarify.
Answer: Sorry for the mistake, it was corrected in the revision that the residual ZWD is modeled as a random walk.

Page 12, line 208: Please, comment use of 2-hour piece-wise constant in ZWD modeling since it is rather long interval for using constant value for modeling the troposphere and may cause the degradation in performance.
Answer: Sorry for the mistake, it was removed.

Page 12: What was a relative weighting (if any) applied for phase and code observations for different systems?
Answer: "The a priori noise value of 2 mm for the phase raw observables and 0.6 m for the code raw observables are applied for each system." was added.

Page 12: Which elevation-dependent weighting function was used?
Answer: We added: "an elevation-dependent weighting ($e < 30°$, 2*sin ($e$); otherwise 1)".

Page 12, line 219: add the reference to IGS tropo products (Byram et al., ION, 2011).
Answer: This reference was cited.

Page 13, line 227: ECMWF ZTDs show good agreement with the IGS ZTDs (plural otherwise shows)
Answer: Corrected.

Page 13, lines 245-250: A strong latitudinal dependency of ZTDs from the MetOffice UK global model with respect to GPS ZTDs was described in Dousa J and Bennitt G, GPS solutions, 2012.
Answer: "Similar findings were demonstrated in Dousa J and Bennitt G (2013), where a strong latitudinal dependency of ZTDs from the UK Met Office global model with respect to GPS ZTDs was described." was added in the revised manuscripts. This reference was also added to the reference list.

Pages 15-16 + Fig 7: How were percentages 32%, 37.5%, 25% derived – supposed the values are calculated as representative over all the stations, but currently it seems they are reported for the station WIND (figure). Anyways, please provide more details about the calculation of statistics (i.e. underlying their representativeness) in terms of

number of data/stations used (included/excluded), criteria applied for achieving the convergence, 'centimetre/decimetre-level' accuracy etc.

Answer: The percentages 32%, 37.5%, 25% were obtained for station WIND, while the averaged results from all four-system stations (shown in Figure 1) were displayed in Figure 8. The percentage of "20.0%, 32.0%, and 25.0%" achieved from the statistical results was updated as the representative over all the stations in the revision. A positioning accuracy less than 10 cm was defined as cm-level, and that of several dm was defined as dm-level. The convergence time was defined as the time required for the horizontal components to be better than 10 cm, and the one needed for the vertical component to be better than 20 cm.

Page 17, line 306: … before the convergence. (the article)
Answer: Corrected.

Page 17, line 305: … position accuracy … It should be also explained which reference values were used in the accuracy assessment. Here, I suppose the 'position accuracy' should mean 'North component accuracy' since East & Up follows and no more North.
Answer: We added that: "The post-processing weekly solution is used as the reference position." Yes, we updated the text to "The positioning accuracy for the north component" for clarification.

Page 18, line 340: about:17 (add space)
Answer: Added.

Page 19, line 366: … before the convergence (the article)
Answer: Corrected.

Page 20, line 367: … 2.5%, 12.1% and 18.7% - only here it is mentioned the values are gained 'after' the reaching convergence (line 366). How the convergence time is determined? Neither abstract nor Section 3.2 mentioned it concerns the period after reaching the convergence.
Answer: We added that: "The convergence time was defined as the period after reaching an accuracy of better than 10 cm for the horizontal components, and that of better than 20 cm for the vertical component."